# Characteristics and predictors of out-of-hospital cardiac arrest in young adults hospitalized with acute coronary syndrome: A retrospective cohort study of 30,000 patients in the Gulf region

Abdulelah H. Alsaeed[1], Ahmed Hersi[1], Tarek Kashour[1], Mohammad Zubaid[2], Jassim Al Suwaidi[3], Haitham Amin[4], Wael AlMahmeed[5], Kadhim Sulaiman[6], Ahmed Al-Motarreb[7], Khalid F. Alhabib[1], Wael Alqarawi[1,8] *

1 Department of Cardiac Sciences, College of Medicine, King Saud University, Riyadh, Saudi Arabia, 2 Department of Medicine, Faculty of Medicine, Kuwait University, Kuwait City, Kuwait, 3 Heart Hospital, Hamad Medical Corporation, Doha, Qatar, 4 Mohammed Bin Khalifa Cardiac Centre, Awali, Bahrain, 5 Heart and Vascular Institute, Cleveland Clinic Abu Dhabi, Al Maryah Island, Abu Dhabi, United Arab Emirates, 6 Department of Cardiology, Royal Hospital, Muscat, Oman, 7 Internal Medicine Department, Faculty of Medicine, Sana'a University, Sana'a, Yemen, 8 University of Ottawa Heart Institute, University of Ottawa, Ottawa, Ontario, Canada

* Walqarawi@ksu.edu.sa

## Abstract

### Introduction

The characteristics of young adults with out-of-hospital cardiac arrest (OHCA) due to acute coronary syndrome (ACS) has not been well described. The mean age of gulf citizens in ACS registries is 10–15 years younger than their western counterparts, which provided us with a unique opportunity to investigate the characteristics and predictors of OHCA in young adults presenting with ACS.

### Methodology

This was a retrospective cohort study using data from 7 prospective ACS registries in the Gulf region. In brief, all registries included consecutive adults who were admitted with ACS. OHCA was defined as cardiac arrest upon presentation (i.e., before admission to the hospital). We described the characteristics of young adults (< 50 years) who had OHCA and performed multivariate logistic regression analysis to assess independent predictors of OHCA.

### Results

A total of 31,620 ACS patients were included in the study. There were 611 (1.93%) OHCA cases in the whole cohort [188/10,848 (1.73%) in young adults vs 423/20,772 (2.04%) in older adults, p = 0.06]. Young adults were predominantly males presenting with ST-elevation myocardial infarction (STEMI) [182/188 (96.8%) and 172/188 (91.49%), respectively]. OHCA was the sentinel event of coronary artery disease (CAD) in 70% of young adults.

**Data Availability Statement:** All dataset files are available from the figshare.com database (DOI: https://dx.doi.org/10.6084/m9.figshare.22340509).

**Funding:** A.A received the Prof. Mohammed Al-Faqih research award as part of the 33rd annual conference of the Saudi Heart Association. Found at: https://saudi-heart.com The funders had no role in study design, data collection and analysis, decision to publish, or preparation of the manuscript.

**Competing interests:** The authors have declared that no competing interests exist.

STEMI, male sex, and non-smoking status were found to be independent predictors of OHCA [OR = 5.862 (95% CI 2.623–13.096), OR: 4.515 (95% CI 1.085–18.786), and OR = 2.27 (95% CI 1.335–3.86), respectively].

## Conclusion

We observed a lower prevalence of OHCA in ACS patients in our region as compared to previous literature from other regions. Moreover, OHCA was the sentinel event of CAD in the majority of young adults, who were predominantly males with STEMIs. These findings should help risk-stratify patients with ACS and inform further research into the characteristics of OHCA in young adults.

## Introduction

Out-of-hospital cardiac arrest (OHCA) is the sudden cessation of cardiac mechanical contractility which is most commonly caused by coronary artery disease (CAD) [1]. While previous studies have investigated the characteristics and predictors of OHCA with acute coronary syndrome (ACS) in general, little is known about young adults, given their limited numbers in those studies [2–6]. The average age of ACS patients in the Gulf countries is 10–15 years younger than their western counterparts [7–13]. This provided us with a unique opportunity to investigate OHCA in young adults with ACS. Not only that this will help us better understand the distinct features and predictors of OHCA in young adults, but also will inform us about the characteristics of CAD as a cause of sudden cardiac death (SCD) in this age group. Because SCD is the most common consequence of OHCA and due to the systematic lack of post-mortem autopsy in our region, this cohort of patients can give us important insights into the characteristics of CAD as a cause of SCD in young adults, notwithstanding survival bias [14, 15]. As such, we sought to describe the characteristics and predictors of OHCA in young adults hospitalized with ACS.

## Methodology

### Data source

Data for this study were obtained by combining 7 prospective ACS registries in the Gulf region [7–13]. These registries have similar methodology with few differences. In brief, all registries included consecutive adults who were admitted to public or private hospitals with ACS. ACS and all other data variables were defined based on the American College of Cardiology clinical data standards [16]. The recruitment period extended from 2005 to 2017 across all registries. One-year follow-up data was available for 4 out of the 7 registries. Details about each registry are depicted in S1 Table [17].

### Variables

We included demographics, medical history, presentation data, reperfusion therapy details, in-hospital and discharge medications, in-hospital complications, and 30-day and 1-year mortality. Young age was defined as age of 50 or less, based on previous literature suggesting that causes of OHCA/SCD are distinctly different in those younger than 50 where CAD is not the most common cause [18]. OHCA was defined as cardiac arrest upon presentation (i.e., before admission to the hospital).

## Ethical considerations

The study was exempted from the submission for review by the Institutional Review Board (IRB), College of Medicine, King Saud University, Riyadh, Saudi Arabia. As the data used in this study consisted of secondary published data that has been completely de-identified with no direct or indirect identifiers available in the database and from the registries that were individually approved through the authors by their respective IRB institutions as follows: [7–13]

RACE 1, RACE 2, RACE 3, STARS, and SPACE: The registries were approved by the IRB of College of Medicine, King Saud University, Riyadh, Saudi Arabia, with additional approval by each participating hospital.

COAST: The registry was approved by the institutional review board of each participating hospital (29 hospitals) in each of the following countries: Kuwait, Bahrain, Oman, and the United Arabic Emirates.

REPERFUSE: The registry was approved by the Kuwait Ministry of Health's ethics committee for the protection of human subjects.

## Statistical analysis

Continuous data were reported as mean ($\pm$standard deviation) and categorical data as numbers (percentages). Student's t-test, Chi Square test and Fisher's exact test were used when appropriate to analyze data. Given that statistical significance can be misleading in the presence of a large sample size, we reported the effect size using the standardized mean difference for continuous variables and phi coefficient for categorical variables. Ten percent or less ($< 10\%$) is considered a small effect size for both statistics [19, 20].

We performed sensitivity analysis including time-dependent analysis, with a cutoff year of 2011. This was done as the study had a long recruitment period extending from 2005 to 2017. Additional analysis for "very young" patients was performed, where the age cutoff was <40 years instead of <50 years given that CAD was found to be the main cause of OHCA in some studies [14].

To assess the independent predictors of OHCA, we performed multivariate logistic regression modeling to adjust for important confounders. We included all baseline characteristics in the model. Moreover, we performed univariate and multivariate logistic regression analysis for potential predictors of in-hospital and 1-year mortality among the entire ACS sample. Analyses were performed using SAS (version *9.4*; The *SAS* Institute, USA) and P values of $< 0.05$ were considered.

## Results

### Overall cohort

A total of 31,620 ACS patients were included in the study, with 10,848 (34.31%) of them being young ($\leq$50 years of age). There were 611 (1.93%) OHCA cases in the whole cohort. The mean age of OHCA patients in total was 58. $\pm$ 14, with 489/611 (80.03%) of them being males. Overall, 346/611 (56.63%) of OHCA patients survived and were discharged from the hospital.

### OHCA patients (young vs old)

Table 1 shows the characteristics of young adults with OHCA as compared to older adults. The prevalence of OHCA was comparable between young and older adults [188/10848 (1.73%) in

**Table 1. Characteristics of patients with out-of-hospital cardiac arrest (young vs older adults).**

| Variables | Young | Old | Total | P value* | ES |
|---|---|---|---|---|---|
|  | (N = 188) | (N = 423) | (N = 611) |  |  |
| **Demographics** |  |  |  |  |  |
| Age | 42 ± 6 | 65 ± 10 | 58 ± 14 | < .0001 | -2.4122 |
| Sex (male) | 182/188 | 307/423 | 489/611 | < .0001 | -0.2798 |
|  | (96.81%) | (72.58%) | (80.03%) |  |  |
| Body mass index | 28 ± 5 | 28 ± 6 | 28 ± 6 | 0.4221 | -0.0676 |
| Ethnicity (Arab) | 81/188 | 293/423 | 374/611 | < .0001 | 0.2480 |
|  | (43.09%) | (69.27%) | (61.21%) |  |  |
| **Medical history** |  |  |  |  |  |
| Diabetes mellitus | 48/188 | 218/422 | 266/610 | < .0001 | 0.2433 |
|  | (25.53%) | (51.66%) | (43.61%) |  |  |
| Hypertension | 41/188 | 250/423 | 291/611 | < .0001 | 0.3446 |
|  | (21.81%) | (59.10%) | (47.63%) |  |  |
| Hyperlipidemia | 37/186 | 160/416 | 197/602 | < .0001 | 0.1829 |
|  | (19.89%) | (38.46%) | (32.72%) |  |  |
| MI or angina | 40/188 | 164/423 | 204/611 | < .0001 | 0.1712 |
|  | (21.28%) | (38.77%) | (33.39%) |  |  |
| Heart failure | 2/108 | 48/236 | 50/344 | < .0001 | 0.2434 |
|  | (1.85%) | (20.34%) | (14.53%) |  |  |
| Chronic renal failure | 2/183 | 48/408 | 50/591 | < .0001 | 0.1773 |
|  | (1.09%) | (11.76%) | (8.46%) |  |  |
| Smoking status (current smokers) | 65/188 | 79/423 | 144/611 | < .0001 | -0.1729 |
|  | (34.57%) | (18.68%) | (23.57%) |  |  |
| **Presentation data** |  |  |  |  |  |
| Grace score | 158 ± 46 | 206 ± 53 | 195 ± 55 | < .0001 | -0.9395 |
| Arrival by ambulance | 55/188 | 140/421 | 195/609 | 0.3285 | 0.0396 |
|  | (29.26%) | (33.25%) | (32.02%) |  |  |
| Presentation Killip class (Killip class 1) | 99/187 | 162/423 | 261/610 | 0.0008 | -0.1365 |
|  | (52.94%) | (38.30%) | (42.79%) |  |  |
| Type of MI (STEMI) | 172/188 | 304/422 | 476/610 | < .0001 | -0.2169 |
|  | (91.49%) | (72.04%) | (78.03%) |  |  |
| Cardiac arrest as the sentinel event of CAD | 29/41 | 61/135 | 90/176 | 0.0042 | -0.2160 |
|  | (70.73%) | (54.81%) | (51.14%) |  |  |
| LV function in Echo (normal) | 23/157 | 56/317 | 79/474 | 0.4070 | 0.0381 |
|  | (14.65%) | (17.67%) | (16.67%) |  |  |
| **Reperfusion therapy details** |  |  |  |  |  |
| Symptoms to hospital arrival time | 123 | 121 | 125 | 0.0033 | -0.2438 |
|  | (IQR: 222)** | (IQR: 125)** | (IQR: 210)** |  |  |
| Primary PCI in STEMI patients | 56/172 | 68/304 | 124/476 | 0.0150 | -0.1115 |
|  | (32.56%) | (22.37%) | (26.05%) |  |  |
| CABG | 1/174 | 7/334 | 8/508 | 0.1913 | 0.0580 |
|  | (0.57%) | (2.10%) | (1.57%) |  |  |
| STEMI thrombolytic therapy | 81/167 | 127/298 | 208/465 | 0.2207 | -0.0568 |
|  | (48.50%) | (42.62%) | (44.73%) |  |  |
| **In hospital medication** |  |  |  |  |  |
| Aspirin | 177/188 | 395/422 | 572/610 | 0.7963 | -0.0105 |
|  | (94.15%) | (93.60%) | (93.77%) |  |  |

(*Continued*)

**Table 1.** (Continued)

| Variables | Young | Old | Total | P value* | ES |
|---|---|---|---|---|---|
| | (N = 188) | (N = 423) | (N = 611) | | |
| GP 2b/3a inhibitors | 40/188 | 60/422 | 100/610 | 0.0297 | -0.0880 |
| | (21.28%) | (14.22%) | (16.39%) | | |
| Other antiplatelets | 153/188 | 301/423 | 454/611 | 0.0076 | -0.1080 |
| | (81.38%) | (71.16%) | (74.30%) | | |
| Heparins (UH or LMWH) | 162/188 | 374/422 | 536/610 | 0.3911 | 0.0347 |
| | (86.17%) | (88.63%) | (87.87%) | | |
| Beta blockers | 88/188 | 145/421 | 233/609 | 0.0037 | -0.1175 |
| | (46.81%) | (34.44%) | (38.26%) | | |
| ACE-I or ARB | 85/188 | 163/423 | 248/611 | 0.1208 | -0.0628 |
| | (45.21%) | (38.53%) | (40.59%) | | |
| Statin | 150/188 | 350/422 | 500/610 | 0.3499 | 0.0378 |
| | (79.79%) | (82.94%) | (81.97%) | | |
| **In hospital course** | | | | | |
| Elective PCI | 18/177 | 23/330 | 41/507 | 0.2078 | -0.0559 |
| | (10.17%) | (6.97%) | (8.09%) | | |
| Elective coronary angiogram | 40/160 | 54/287 | 94/447 | 0.1240 | -0.0728 |
| | (25.00%) | (18.82%) | (21.03%) | | |
| **In hospital complications** | | | | | |
| In-hospital heart failure | 53/188 | 226/423 | 279/611 | < .0001 | 0.2338 |
| | (28.19%) | (53.43%) | (45.66%) | | |
| Recurrent MI (In Hospital Infarction/Re-Infarction) | 9/188 | 30/423 | 39/611 | 0.2820 | 0.0435 |
| | (4.79%) | (7.09%) | (6.38%) | | |
| Stroke | 4/188 | 21/422 | 25/610 | 0.1013 | 0.0663 |
| | (2.13%) | (4.98%) | (4.10%) | | |
| Major Bleeding | 11/188 | 20/423 | 31/611 | 0.5594 | -0.0236 |
| | (5.85%) | (4.73%) | (5.07%) | | |
| **Mortality** | | | | | |
| Mortality in-hospital | 39/188 | 226/423 | 265/611 | < .0001 | 0.3044 |
| | (20.74%) | (53.43%) | (43.37%) | | |
| One month mortality | 11/48 | 118/180 | 129/228 | < .0001 | 0.3507 |
| | (22.92%) | (65.56%) | (56.58%) | | |
| One year mortality | 9/44 | 119/176 | 128/220 | < .0001 | 0.3824 |
| | (20.45%) | (67.61%) | (58.18%) | | |

ES: Effect size. MI: Myocardial infarction. STEMI: ST elevation myocardial infarction. NSTEMI: Non-ST elevation myocardial infarction. LV: Left ventricle. PCI: Percutaneous coronary intervention. CABG: Coronary artery bypass graft surgery. UH: Unfractionated heparin. LWMH: Low molecular weight heparin. IQR: Interquartile range.

*P values were the result of the comparison between young vs older adults.

** These values did not have a normal distribution, and so, median and interquartile range was used instead.

young adults vs 423/20772 (2.04%) in older adults; p = 0.063]. Young adults were predominantly males presenting with STEMI (182/188 (96.8%) and 172/188 (91.49%), respectively). When compared to older adults, they were more likely to be males (182/188 (96.81%) vs 307/423 (72.58%); p < .0001), experience STEMI (172/188 (91.49%) vs 304/422 (72.04%); p < .0001), and have less prevalence of CAD risk factors such as hypertension (41/188 (21.81%) vs 250/423 (59.10%); p < .0001) and diabetes mellitus (48/188 (25.53%) vs 218/422 (51.66%); p <

.0001). In contrast to older adults, cardiac arrest was the sentinel event of CAD (i.e., the presenting symptom) in the majority of young adults (29/41 (70.73%) vs 61/135 (54.81%); p = 0.0.004). Young adults were more likely to receive optimal therapy such as primary PCI (56/172 (32.56%) vs 68/304 (22.37%); p < .0001) and beta blockers (88/188 (46.81%) vs 145/421 (34.44%); p = 0.0037). They also had consistently lower mortality rates including in-hospital mortality [39/188 (20.74%) vs 226/423 (53.43%); p < .0001)], 30-days mortality [(11/48 (22.92%) vs 118/180 (65.56%); p < .0001)] and one-year mortality [(9/44 (20.45%) vs 119/176 (67.61%); p-value < .0001)].

## Young adults (OHCA vs no OHCA)

Table 2 compares young adults who developed OHCA with those who did not. There were no major differences in baseline characteristics although some were statistically significantly different. Young adults who developed OHCA had higher mortality rates including in-hospital mortality [39/188 (20.74%) vs 154/10657 (1.46%); p < .0001] and 30-days mortality [11/48 (22.92%) vs 109/4044 (2.70%); p < .0001].

## Predictors of OHCA

STEMI and non-smoking were shared independent predictors of OHCA between young and older adults [STEMI: OR = 5.862 (95% CI 2.623–13.096) and OR = 4.010 (95% CI 2.861–5.621), respectively. Non-smoking: OR = 2.27 (95% CI 1.335–3.86) and OR = 2.031 (95% CI 1.222–3.374), respectively]. Male sex was only a predictor in young adults (p = 0.0383) and comorbidities were only predictors for older adults [heart failure (p < .0001), and chronic kidney disease (CKD) (p < .0001)]. Table 3 summaries results of the logistic regression for young and older adults and Fig 1 depicts independent predictors.

## Sensitivity analysis

S2 Table shows the characteristics and predictors of "very young" OHCA patients (<40 years). "Very young" OHCA patients had similar features to what we described in young patients (<50 years), with the majority being male [53/54 (98.15%)] and presenting with STEMI [46/54 (85.19%)]. However, "very young" patients had an even lower prevalence of CAD risk factors such as hypertension [4/54 (7.41%)] and diabetes mellitus [6/54 (11.11%)].

S3 and S4 Tables report time-dependent analysis before and after 2011, respectively. The total prevalence of OHCA before 2011 was 262/7,310 (3.58%) which was higher after 2011 [349/3,537 (9.87%)]. However, the proportion of young adults with OHCA remained similar for both time periods [79/262 (30.15%)] before 2011 and [109/349 (31.23%)] after 2011. Differences were also seen regarding reperfusion therapy. For example, the rate of primary PCI in STEMI patients improved over time, from 8.15% before 2011, to 37.33% after 2011.

## Predictors of mortality

S5A, S5B Table show the predictors of in-hospital and 1-year mortality in ACS patients. OHCA was found to be an independent predictor of in-hospital but not 1-year mortality [OR = 2.673 (95% CI 1.271–5.620) for young adults, OR = 3.194 (95% CI 1.872–5.450) for older adults; p < .0001] and [OR = 1.547 (95% CI 0.246–9.744) for young adults, OR = 0.816 (95% CI 0.114–5.823) for older adults; p = 0.6394], respectively. There was no interaction between OHCA and young age for both in-hospital and 1-year mortality [p = 0.6960] and [p = 0.6394], respectively. Revascularization was protective for in-hospital and 1-year mortality [OR = 0.329 (95% CI 0.241–0.448); p < .0001] and [OR = 0.674 (95% CI 0.478–0.948);

**Table 2. Characteristics of young adults (with vs without out-of-hospital cardiac arrest).**

| Variables | Yes | No | Total | P value | ES |
|---|---|---|---|---|---|
| | (N = 188) | (N = 10,660) | (N = 10,848) | | |
| **Demographics** | | | | | |
| Age | 43 ± 6 | 43 ± 6 | 43 ± 6 | 0.0891 | -0.1251 |
| Sex (male) | 182/188 | 9448/10659 | 9630/10847 | 0.0004 | 0.0338 |
| | (96.81%) | (88.64%) | (88.78%) | | |
| Body mass index | 28 ± 5 | 28 ± 6 | 28 ± 6 | 0.6779 | -0.0316 |
| Ethnicity (Arab) | 81/188 | 5448/10659 | 5529/10847 | 0.0291 | -0.0210 |
| | (43.09%) | (51.11%) | (50.97%) | | |
| **Medical history** | | | | | |
| Diabetes mellitus | 48/188 | 3513/10615 | 3561/10803 | 0.0288 | -0.0210 |
| | (25.53%) | (33.09%) | (32.96%) | | |
| Hypertension | 41/188 | 3640/10613 | 3681/10801 | 0.0003 | -0.0345 |
| | (21.81%) | (34.30%) | (34.08%) | | |
| Hyperlipidemia | 37/186 | 2905/10096 | 2942/10282 | 0.0079 | -0.0262 |
| | (19.89%) | (28.77%) | (28.61%) | | |
| MI or angina | 40/188 | 2728/10660 | 2768/10848 | 0.1786 | -0.0129 |
| | (21.28%) | (25.59%) | (25.52%) | | |
| Heart failure | 2/108 | 125/5634 | 127/5742 | 0.7974 | -0.0034 |
| | (1.85%) | (2.22%) | (2.21%) | | |
| Chronic renal failure | 2/183 | 88/9025 | 90/9208 | 0.8726 | 0.0017 |
| | (1.09%) | (0.98%) | (0.98%) | | |
| Smoking status (current smokers) | 65/188 | 3899/10648 | 3964/10836 | 0.5643 | -0.0055 |
| | (34.57%) | (36.62%) | (36.58%) | | |
| **Presentation data** | | | | | |
| Grace score | 158 ± 46 | 91 ± 29 | 92 ± 30 | < .0001 | 2.29148 |
| Arrival by ambulance | 55/188 | 2066/10503 | 2121/10691 | 0.0011 | 0.0316 |
| | (29.26%) | (19.67%) | (19.84%) | | |
| Presentation Killip class (Killip class 1) | 99/187 | 9319/10497 | 9418/10684 | < .0001 | -0.1454 |
| | (52.94%) | (88.78%) | (88.15%) | | |
| Type of MI (STEMI) | 172/188 | 6313/10653 | 6485/10841 | < .0001 | 0.0858 |
| | (91.49%) | (59.26%) | (59.82%) | | |
| LV function in Echo (normal) | 23/157 | 3196/9193 | 3219/9350 | < .0001 | -0.0544 |
| | (14.65%) | (34.77%) | (34.43%) | | |
| **Reperfusion therapy details** | | | | | |
| Symptoms to hospital arrival time | 121 | 165 | 165 | < .0001 | -0.2378 |
| | (IQR: 125)* | (IQR: 274)* | (IQR: 267)* | | |
| Primary PCI in STEMI patients | 56/172 | 1362/6313 | 1418/6485 | 0.0006 | 0.0427 |
| | (32.56%) | (21.57%) | (21.87%) | | |
| CABG | 1/174 | 235/9742 | 236/9916 | 0.1150 | -0.0158 |
| | (0.57%) | (2.41%) | (2.38%) | | |
| STEMI thrombolytic therapy | 81/167 | 3426/6208 | 3507/6375 | 0.0866 | -0.0215 |
| | (48.50%) | (55.19%) | (55.01%) | | |
| **In hospital medication** | | | | | |
| Aspirin | 177/188 | 10523/10651 | 10700/10839 | < .0001 | -0.0539 |
| | (94.15%) | (98.80%) | (98.72%) | | |

(*Continued*)

**Table 2.** (Continued)

| Variables | Yes | No | Total | P value | ES |
|---|---|---|---|---|---|
|  | **(N = 188)** | **(N = 10,660)** | **(N = 10,848)** |  |  |
| GP 2b/3a inhibitors | 40/188 | 1961/10653 | 2001/10841 | 0.3149 | 0.0097 |
|  | (21.28%) | (18.41%) | (18.46%) |  |  |
| Other antiplatelets | 153/188 | 8470/10660 | 8623/10848 | 0.5165 | 0.0062 |
|  | (81.38%) | (79.46%) | (79.49%) |  |  |
| Heparins (UH or LMWH) | 162/188 | 9395/10658 | 9557/10846 | 0.4057 | -0.0080 |
|  | (86.17%) | (88.15%) | (88.12%) |  |  |
| Beta blockers | 88/188 | 8296/10646 | 8384/10834 | < .0001 | -0.0971 |
|  | (46.81%) | (77.93%) | (77.39%) |  |  |
| ACE-I or ARB | 85/188 | 7534/10657 | 7619/10845 | < .0001 | -0.0728 |
|  | (45.21%) | (70.70%) | (70.25%) |  |  |
| Statin | 150/188 | 10079/10650 | 10229/10838 | < .0001 | -0.0842 |
|  | (79.79%) | (94.64%) | (94.38%) |  |  |
| **In hospital course** |  |  |  |  |  |
| Elective PCI | 18/177 | 1218/9917 | 1236/10094 | 0.3954 | -0.0085 |
|  | (10.17%) | (12.28%) | (12.24%) |  |  |
| Elective coronary angiogram | 40/160 | 3293/9443 | 3333/9603 | 0.0093 | -0.0265 |
|  | (25.00%) | (34.87%) | (34.71%) |  |  |
| **In hospital complications** |  |  |  |  |  |
| In-hospital heart failure | 53/188 | 729/10649 | 782/10837 | < .0001 | 0.1077 |
|  | (28.19%) | (6.85%) | (7.22%) |  |  |
| Recurrent MI (In Hospital Infarction/Re-Infarction) | 9/188 | 170/10650 | 179/10838 | 0.0007 | 0.0327 |
|  | (4.79%) | (1.60%) | (1.65%) |  |  |
| Stroke | 4/188 | 43/10631 | 47/10819 | 0.0004 | 0.0342 |
|  | (2.13%) | (0.40%) | (0.43%) |  |  |
| Major Bleeding | 11/188 | 36/10654 | 47/10842 | < .0001 | 0.1095 |
|  | (5.85%) | (0.34%) | (0.43%) |  |  |
| **Mortality** |  |  |  |  |  |
| Mortality in-hospital | 39/188 | 154/10657 | 193/10845 | < .0001 | 0.1905 |
|  | (20.74%) | (1.46%) | (1.78%) |  |  |
| One month mortality | 11/48 | 109/4044 | 120/4092 | < .0001 | 0.1290 |
|  | (22.92%) | (2.70%) | (2.93%) |  |  |
| One year mortality | 9/44 | 148/3661 | 157/3705 | < .0001 | 0.0883 |
|  | (20.45%) | (4.04%) | (4.24%) |  |  |

ES: Effect size. MI: Myocardial infarction. STEMI: ST elevation myocardial infarction. NSTEMI: Non-ST elevation myocardial infarction. LV: Left ventricle. PCI: Percutaneous coronary intervention. CABG: Coronary artery bypass graft surgery. UH: Unfractionated heparin. LWMH: Low molecular weight heparin. IQR: Interquartile range

* These values did not have a normal distribution, and so, median and interquartile range was used instead.

p = 0.0236], respectively. S5C Table compares the mortality rates of OHCA patients before and after 2011. We observed higher mortality before 2011, albeit not statistically significant.

## Discussion

In this registry-based study of OHCA patients with ACS, we found the prevalence of OHCA to be lower than previously reported by other investigators, the characteristics and predictors of OHCA to be distinctly different in young adults when compared to older adults, and cardiac

**Table 3. Multivariate logistic regression for young and older adults.**

| Variables | YOUNG ADULTS | | | OLDER ADULTS | | |
|---|---|---|---|---|---|---|
| | OR | 95% CI | P value | OR | 95% CI | P value |
| Age | 0.984 | 0.954–1.015 | 0.2960 | **1.026** | **1.011–1.040** | **0.0005** |
| Sex (Male) | **4.515** | **1.085–18.786** | **0.0383** | 0.891 | 0.651–1.219 | 0.4713 |
| Ethnicity (Arab) | 0.925 | 0.616–1.390 | 0.7070 | 0.724 | 0.517–1.016 | 0.0617 |
| Diabetes mellitus | 1.227 | 0.793–1.899 | 0.3581 | 1.010 | 0.755–1.351 | 0.9482 |
| Hypertension | 0.636 | 0.382–1.059 | 0.0818 | 0.847 | 0.622–1.153 | 0.2916 |
| Hyperlipidemia | 0.667 | 0.394–1.130 | 0.1321 | 0.928 | 0.684–1.258 | 0.6291 |
| Coronary artery disease | 1.286 | 0.741–2.231 | 0.3718 | 0.778 | 0.557–1.086 | 0.1404 |
| History of stroke | 1.934 | 0.425–8.806 | 0.3938 | 1.320 | 0.832–2.094 | 0.2391 |
| History of CHF | 1.746 | 0.356–8.549 | 0.4919 | **3.336** | **2.226–4.999** | **< .0001** |
| History of CKD | 1.044 | 0.118–9.204 | 0.9690 | **3.093** | **2.057–4.652** | **< .0001** |
| Smoking | **0.441** | **0.259–0.749** | **0.0025** | **0.492** | **0.296–0.818** | **0.0063** |
| STEMI | **5.862** | **2.623–13.096** | **< .0001** | **4.010** | **2.861–5.621** | **< .0001** |

OR: Odds ratio. CI: Confidence interval. CHF: Congestive heart failure. CKD: Chronic kidney disease. STEMI: ST elevation myocardial infarction.

arrest to be the sentinel event of CAD in the majority of young adults. These findings will hopefully inform clinical care of these patients and highlight research gaps for future studies.

The lower prevalence of OHCA in our study as compared to previous literature is interesting and is worth a closer look. Previous studies have reported a prevalence of 5–11%, which is significantly higher than our 1.93% [3–5, 21, 22]. There are several potential explanations for this observation. First, some studies included only STEMI patients, which is shown in our study and others to be an independent predictor of OHCA [3, 5, 21]. Second, the use of emergency medical services (EMS) was low in our study (only 32% of OHCA patients in our study arrived by ambulance). Poor utilization of EMS services in ACS cases, especially ones that were complicated by OHCA, is associated with higher mortality [23, 24] as the odds of survival decreases 7–10% each minute defibrillation is delayed [24]. It is possible that a significant proportion of patients with OHCA went on to have SCD and were never included in these registries. Third, the lower prevalence of OHCA might be related to the "smoking paradox" [25].

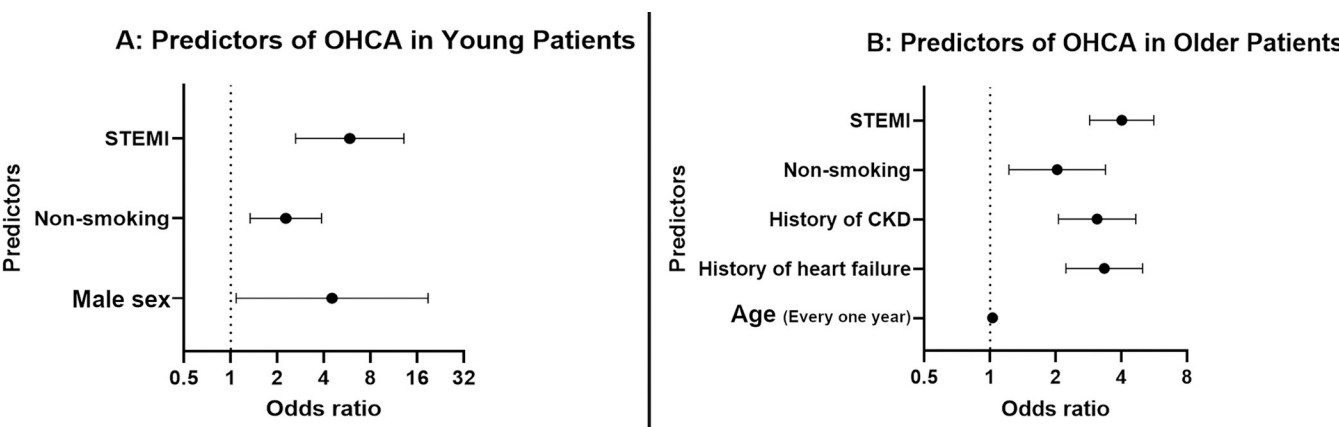

**Fig 1. Predictors of OHCA in young and older adults.** STEMI: ST elevation myocardial infarction. CKD: Chronic kidney disease. 1A: Predictors of OHCA in Young Patients and 1B: Predictors of OHCA in Older Patients.

"Smoking paradox" refers to the observation found by many studies that smokers with cardiovascular diseases including ACS have better outcomes than non-smokers [25–27]. It is unclear if "smoking paradox" represents a true phenomenon or a spurious finding. Aune et al reported a systematic review that reviewed the evidence behind "smoking paradox" [25]. They examined 17 studies in total and found that this paradox is not always found, especially in studies using contemporary therapies. Nonetheless, a more recent study utilizing a prospective cohort study of more than 37,000 patients with myocardial infarction found that smokers are younger and have fewer comorbidities [26]. However, even after adjustment for confounders, smokers were found to have a lower in-hospital mortality. It is certainly possible that residual confounding explains this apparent association in this study and others, including ours. While the exact mechanism of this paradox is not clear, our study also confirmed the protective effect of smoking on developing OHCA in both young and older adults [smoking: OR = 0.441 (95% CI 0.259–0.749) and OR = 0.492 (95% CI 0.296–0.818), respectively]. Smoking is quite prevalent in the Gulf region with a rate that is 1.5–2 times higher than other registries, which might be a factor to explain the lower prevalence of OHCA overall [17, 28, 29]. Last, age has been inconsistently shown to be associated with OHCA in ACS patients, and could be another contributing factor [2, 3, 5].

Similar to previous studies, summarized in Table 4, we found that STEMI [2, 4], age [2, 3, 5], history of heart failure [2–4], and CKD [2–6] are predictors of OHCA in ACS patients. Our study is unique in that it examined young adults separately, which yielded 2 important findings with clinical implications. First, we found that young adults were predominantly males who presented with STEMI. In fact, only 1 young patient (1/188, 0.5%) had OHCA without these 2 predictors (a female patient with NSTEMI). This is helpful in risk-stratifying young adults with ACS for OHCA where these criteria can be used to select higher risk patients to examine future interventions and to prioritize them for more urgent care. Second, the rate of cardiac arrest being the sentinel event of CAD was distinctly different in young versus older adults (29/41 70.73%) vs 61/135 54.81%); p = 0.004). It is important to note that previous literature suggested that the prevalence of SCA/SCD being the sentinel event is around 50%, which is similar to older adults in our study [30, 31]. However, no study to our knowledge examined this in the young population [30, 31]. The fact that 70% of young adults had OHCA as the sentinel event of CAD emphasizes the importance of medical autopsy and the potential harm and anxiety of assuming inherited arrhythmic conditions to be the cause of SCD when approaching families with a SCD at young age, as suggested by guidelines [32]. In addition, our finding that the majority of young adults had OHCA due to ACS without known cardiovascular risk factors should deter the use of risk factors and comorbidities as a guide to whom should get a medical autopsy in the case of SCD at young age. Moreover, early screening of CAD, independent of risk factors, needs to be studied in our population to examine its benefit in preventing SCD. It should be noted that this finding is limited by the smaller sample size available for this variable, given that it was not collected in all registries. Further studies with a focus on the presentation of OHCA as the sentinel event of CAD in young patients using a larger sample size need to be performed in order to confirm our findings.

To our knowledge, our study is the first study that investigated OHCA in young ACS patients using a large sample size. However, it has several limitations. First, we included patients at different years ranging from 2005–2017, which might have affected care received. Nonetheless, this should not have a major effect on the characteristics and predictors of OHCA that were the focus of this manuscript. We also performed time-dependent analysis (shown in S3 and S4 Tables), in order to provide data across different time periods. Second, there might be patients with OHCA due to ACS that were not included in these registries if the cause of OHCA was not clearly due to ACS at the time of presentation, a limitation shared

**Table 4. Comparison of the characteristics and predictors of cardiac arrest patients across different studies.**

| Study name/year | Prevalence of OHCA (Survival rate)* | Sample size (OHCA Cases) | Age | STEMI % | Smoker % | Predictors of OHCA | Comment |
|---|---|---|---|---|---|---|---|
| This study | **Overall:** 1.93% (56.63%) **Young:** 1.73% (79.26%) **Older:** 2.04% (46.57%) | 31,620 (611) | **Overall:** 58 ± 14 **Young:** 43 ± 6 **Older:** 65 ± 10 | **Overall:** 78.03% **Young:** 91.49% **Older:** 72.04% | **Overall:** 23.57% **Young:** 34.57% **Older:** 18.68% | **Young:** 1. Male 2. STEMI 3. Non-Smoking **Older:** 1. Age 2. STEMI 3. Non-Smoking 4. History of HF 5. History of CKD | The first major study that investigated OHCA in young ACS patients using a large sample size. |
| Kontos et al.,[3] 2015 | 7.5% (71.2%) | 49,279 (3,716) | **Median:** 61 **IQR:** 52–70 | STEMI only | 45.3% (Current/recent) | 1. Male 2. History of HF 3. Higher creatinine 4. Lack of traditional coronary disease risk 5. No previous Hx of prior MI 6. No previous Hx of prior revascularization | STEMI only. Univariate analysis for associated factors. |
| Dawson et al.,[5] 2020 | 8.4% (63%)** | 12,637 (1,057) | 61 ± 12.5 | STEMI only | 63.7% (Ever smoker) | 1. Younger age 2. Male 3. Absence of dyslipidemia 4. Prior valve surgery 5. Multi-vessel disease 6. LAD culprit 7. Small vessel (< = 2.5 mm) 8. Renal impairment | STEMI only. |
| Kosugi et al.,[6] 2020 | 29.38% (62%) | 480 (141) | **Median:** 63 **IQR:** 52–71 | 76% | 38% | 1. Younger age 2. Non use of calcium-channel antagonists 3. Lower eGFR, per 10 ml/min/1.73 m2 4. Peak CK-MB, per 102 U/l 5. LAD culprit 6. Chronic total occlusion | Single center study with a limited sample (480 total MI patients, 141 OHCA) and it handles most of the OHCA cases in their city. Univariate analysis for associated factors. |
| Jabbari et al.,[21] 2014 | 9.6% (73%) | 1901 (182) | **Median:** 59 **IQR:** 53–68 | 83% | 54% | 1. Younger age 2. Alcohol use >7 units/week 3. Atrial fibrillation 4. FH of sudden death 5. Anterior Infarct location 6. TIMI flow grade 0 before primary PCI 7. Statins before STEMI 8. No previous Hx of angina | Included OHCA and those who had VF during admission but before PCI. STEMI only. |

*(Continued)*

**Table 4.** (Continued)

| Study name/year | Prevalence of OHCA (Survival rate)* | Sample size (OHCA Cases) | Age | STEMI % | Smoker % | Predictors of OHCA | Comment |
|---|---|---|---|---|---|---|---|
| DeFilippis *et al.,*[22] 2018 | 4.8% (79%) | 2097 (100) | 44±5 | 77.0% | 62.0% (Did not specify if current or ever) | 1. STEMI<br>2. Higher creatinine<br>3. Higher troponin<br>4. Smoking<br>5. Cocaine or marijuana use<br>6. No previous Hx of DM<br>7. lower ejection fraction<br>8. Lower ASCVD score | Included only young MI patients (≤50). Univariate analysis for associated factors. |
| Fordyce *et al.,*[2] 2016 | 1.2% (56.6%) | 54,860 (641) | **Median:** 73 **IQR:** 68–79 | 63.2% | 20.4% (Current/ recent smoker) | 1. Younger<br>2. Male<br>3. Smokers<br>4. Atrial fibrillation or flutter within 2 weeks before<br>5. admission<br>6. STEMI<br>7. Cardiogenic shock<br>8. No previous Hx of diabetes<br>9. No previous Hx of HF<br>10. No previous Hx of PAD<br>11. No previous Hx of chronic lung disease<br>12. No previous Hx of prior revascularization | Only studied MI patients 65 years or above. Also, only patients discharged alive were included. Univariate analysis for associated factors. |

OHCA: Out of hospital cardiac arrest. HF: Heart failure. CKD: Chronic kidney disease. PAD: Peripheral arterial disease. ACS: Acute coronary syndrome. MI: Myocardial infarction. STEMI: ST elevation myocardial infarction. NSTEMI: Non-ST elevation myocardial infarction. LV: Left ventricle. LAD: Left anterior descending artery. ASCVD: Atherosclerotic cardiovascular disease. VF: Ventricular fibrillation. PCI: Percutaneous coronary intervention. DM: Diabetes mellitus

*Survival rate refers to survival to hospital discharge.

**30 days survival was used instead of survival to hospital discharge

with all previous similar studies. However, we did not see any evidence of bias toward patients with known CAD or comorbidities. Third, the apparent high survival rate should be interpreted with cation given the survival bias. Indeed, only patients with ACS who survived to hospital admission were included in these registries which results in a biased estimate of survival. Fourth, our finding of OHCA as the sentinel event of CAD was based on a smaller sample compared to the other variables, which affects the precision of this estimate. Last, patients with SCD might have different characteristics than those who survive OHCA. As such, caution should be taken when extrapolating our findings to the SCD population. Yet, our findings shed some light on this largely un-studied population in our region and other regions that share the systematic lack of medical autopsy for SCD.

## Conclusion

This large registry-based study found that young OHCA patients with ACS were more likely to be male and have STEMI with low prevalence of traditional cardiovascular risk factors. Moreover, OHCA was the sentinel event of CAD in many young adults. These findings should inform risk stratification of young adults with ACS and the evaluation of SCD in our region.

## Supporting information

**S1 Table. Details of the registries included in the study.** ACS: Acute coronary syndrome. Gulf COAST: Gulf Locals with Acute Coronary Syndrome Events Registry. Gulf RACE: Gulf Registry of Acute Coronary Events. KSA: Kingdom of Saudi Arabia. Kuwait REPERFUSE: Reperfusion in ST-Segment–Elevation Myocardial Infarction. NSTEMI: non–ST-segment–elevation myocardial infarction. SPACE: Saudi Project for Assessment of Coronary Events. STARS: Saudi Acute Myocardial Infarction Registry. STEMI: ST-segment–elevation myocardial infarction. UA: Unstable angina. UAE: United Arab Emirates.
(DOCX)

**S2 Table. S2A Table. Characteristics of patients with out-of-hospital cardiac arrest [very young (<40 years) vs older adults].** ES: Effect size. MI: Myocardial infarction. STEMI: ST elevation myocardial infarction. NSTEMI: Non-ST elevation myocardial infarction. LV: Left ventricle. PCI: Percutaneous coronary intervention. CABG: Coronary artery bypass graft surgery. UH: Unfractionated heparin. LWMH: Low molecular weight heparin. *P values were the result of the comparison between very young (<40 years) vs older adults. **S2B Table. Characteristics of very young (<40 years) adults (with vs without out-of-hospital cardiac arrest).** ES: Effect size. MI: Myocardial infarction. STEMI: ST elevation myocardial infarction. NSTEMI: Non-ST elevation myocardial infarction. LV: Left ventricle. PCI: Percutaneous coronary intervention. CABG: Coronary artery bypass graft surgery. UH: Unfractionated heparin. LWMH: Low molecular weight heparin.
(DOCX)

**S3 Table. S3A Table Characteristics of patients with out-of-hospital cardiac arrest (young vs older adults) before 2011.** ES: Effect size. MI: Myocardial infarction. STEMI: ST elevation myocardial infarction. NSTEMI: Non-ST elevation myocardial infarction. LV: Left ventricle. PCI: Percutaneous coronary intervention. CABG: Coronary artery bypass graft surgery. UH: Unfractionated heparin. LWMH: Low molecular weight heparin. *P values were the result of the comparison between young vs older adults. **S3B Table. Characteristics of young adults (with vs without out-of-hospital cardiac arrest) before 2011.** ES: Effect size. MI: Myocardial infarction. STEMI: ST elevation myocardial infarction. NSTEMI: Non-ST elevation myocardial infarction. LV: Left ventricle. PCI: Percutaneous coronary intervention. CABG: Coronary artery bypass graft surgery. UH: Unfractionated heparin. LWMH: Low molecular weight heparin.
(DOCX)

**S4 Table. S4A Table Characteristics of patients with out-of-hospital cardiac arrest (young vs older adults) after 2011.** ES: Effect size. MI: Myocardial infarction. STEMI: ST elevation myocardial infarction. NSTEMI: Non-ST elevation myocardial infarction. LV: Left ventricle. PCI: Percutaneous coronary intervention. CABG: Coronary artery bypass graft surgery. UH: Unfractionated heparin. LWMH: Low molecular weight heparin. *P values were the result of the comparison between young vs older adults. **S4B Table. Characteristics of young adults (with vs without out-of-hospital cardiac arrest) after 2011.** ES: Effect size. MI: Myocardial

infarction. STEMI: ST elevation myocardial infarction. NSTEMI: Non-ST elevation myocardial infarction. LV: Left ventricle. PCI: Percutaneous coronary intervention. CABG: Coronary artery bypass graft surgery. UH: Unfractionated heparin. LWMH: Low molecular weight heparin.
(DOCX)

**S5 Table. S5A Table Univariate and multivariate logistic regression for predictors of in-hospital mortality in all ACS patients.** OR: Odds ratio. CI: Confidence interval. MI: Myocardial infarction. STEMI: ST elevation myocardial infarction. LV: Left Ventricle. UH: Unfractionated heparin. LWMH: Low molecular weight heparin. OHCA: Out of hospital cardiac arrest. *Interaction analysis was performed for the variables "young age" and "OHCA" (Young*-OHCA). **S5B Table. Univariate and multivariate logistic regression for predictors of 1-year mortality in all ACS patients.** OR: Odds ratio. CI: Confidence interval. MI: Myocardial infarction. STEMI: ST elevation myocardial infarction. LV: Left Ventricle. UH: Unfractionated heparin. LWMH: Low molecular weight heparin. OHCA: Out of hospital cardiac arrest. *Interaction analysis was performed for the variables "young age" and "OHCA" (Young*-OHCA). **S5C Table. Comparison of mortality rates of OHCA patients before and after 2011.** OHCA: Out of hospital cardiac arrest.
(DOCX)

## Author Contributions

**Conceptualization:** Abdulelah H. Alsaeed, Ahmed Hersi, Tarek Kashour, Mohammad Zubaid, Jassim Al Suwaidi, Haitham Amin, Wael AlMahmeed, Kadhim Sulaiman, Ahmed Al-Motarreb, Khalid F. Alhabib, Wael Alqarawi.

**Formal analysis:** Wael Alqarawi.

**Investigation:** Abdulelah H. Alsaeed, Ahmed Hersi, Tarek Kashour, Mohammad Zubaid, Jassim Al Suwaidi, Haitham Amin, Wael AlMahmeed, Kadhim Sulaiman, Ahmed Al-Motarreb, Khalid F. Alhabib, Wael Alqarawi.

**Methodology:** Abdulelah H. Alsaeed, Ahmed Hersi, Tarek Kashour, Mohammad Zubaid, Jassim Al Suwaidi, Haitham Amin, Wael AlMahmeed, Kadhim Sulaiman, Ahmed Al-Motarreb, Khalid F. Alhabib, Wael Alqarawi.

**Project administration:** Wael Alqarawi.

**Supervision:** Wael Alqarawi.

**Writing – original draft:** Abdulelah H. Alsaeed, Ahmed Hersi, Tarek Kashour, Mohammad Zubaid, Jassim Al Suwaidi, Haitham Amin, Wael AlMahmeed, Kadhim Sulaiman, Ahmed Al-Motarreb, Khalid F. Alhabib, Wael Alqarawi.

**Writing – review & editing:** Abdulelah H. Alsaeed, Ahmed Hersi, Tarek Kashour, Mohammad Zubaid, Jassim Al Suwaidi, Haitham Amin, Wael AlMahmeed, Kadhim Sulaiman, Ahmed Al-Motarreb, Khalid F. Alhabib, Wael Alqarawi.

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
