## [Decision Letter · Decision Letter 0]

19 Jan 2023

PONE-D-22-33714Characteristics and predictors of out-of-hospital cardiac arrest in young adults hospitalized with acute coronary syndrome: A retrospective cohort study of 30,000 patients in the Gulf region.PLOS ONE

Dear Dr. Alqarawi,

Thank you for submitting your manuscript to PLOS ONE. After careful consideration, we feel that it has merit but does not fully meet PLOS ONE’s publication criteria as it currently stands. Therefore, we invite you to submit a revised version of the manuscript that addresses the points raised during the review process.

We look forward to receiving your revised manuscript.

Kind regards,

Shukri AlSaif

Academic Editor

PLOS ONE

Journal Requirements:

2. In the Methods section of your revised manuscript, please include the names of all of the IRBs that approved the seven registries

3. Thank you for including your ethics statement:  "The data used in this study were obtained from 7 registries which were all approved by their respective institution review boards".   

a. For studies reporting research involving human participants, PLOS ONE requires authors to confirm that this specific study was reviewed and approved by an institutional review board (ethics committee) before the study began. Please provide the specific name of the ethics committee/IRB that approved your study, or explain why you did not seek approval in this case.

b. Please provide additional details regarding participant consent. In the ethics statement in the Methods and online submission information, please ensure that you have specified (1) whether consent was informed and (2) what type you obtained (for instance, written or verbal, and if verbal, how it was documented and witnessed). If your study included minors, state whether you obtained consent from parents or guardians. If the need for consent was waived by the ethics committee, please include this information.

Reviewers' comments:

Reviewer's Responses to Questions

**Comments to the Author**

1. Is the manuscript technically sound, and do the data support the conclusions?

Reviewer #1: Yes

Reviewer #2: Yes

2. Has the statistical analysis been performed appropriately and rigorously? 

Reviewer #1: No

Reviewer #2: Yes

3. Have the authors made all data underlying the findings in their manuscript fully available?

Reviewer #1: Yes

Reviewer #2: No

4. Is the manuscript presented in an intelligible fashion and written in standard English?

Reviewer #1: Yes

Reviewer #2: Yes

5. Review Comments to the Author

Reviewer #1: Characteristics and predictors of out-of-hospital cardiac arrest in young adults hospitalized with acute coronary syndrome: A retrospective cohort study of 30,000 patients in the Gulf region.

The authors aim to describe the characteristics and predictors of OHCA in young adults (aged below 50 years) hospitalized with ACS. This is a retrospective cohort study using data from 7 prospective ACS registries in the Gulf region. The main findings are:

- The prevalence of OHCA in an ACS population in the Gulf region is lower as compared to previous literature.

- Young OHCA patients with ACS are more likely to be male and have STEMI with low prevalence of traditional cardiovascular risk factors, compared to older adults with OHCA.

- OHCA is the sentinel event of CAD (i.e., the presenting symptom) in the majority of young adults (29/41 (70.73%)).

Their study is unique because it examines young adults separately. However, there are some remarks to be made on this manuscript.

Major remarks

- It is stated that “causes of OHCA/SCD are distinctly different in those younger than 50 where CAD is not the most common cause.[18]” This is true for patients below 40 years of age, but above 40 also CAD and associated acute coronary syndrome is the main cause of OHCA (Empana et al, JACC, 2022). Please provide data on patients <40 years.

- The third main finding as listed above (OHCA is the sentinel event of CAD in the majority of young adults) is based on a sample size of 41 patients, see page 10 Table 1. Since this is described as one of the main results of the paper, please elaborate further on this small sample size. It is striking that for the other variables in Table 1 page 10 (subset ‘presentation data’) there is an almost complete sample size of 188 OHCA patients.

- Please include this in the future recommendations as well.

- In the statistical analysis section (page 6 line 106) it is described that all variables are reported as means with standard deviations. Are the variables tested for normal distribution? If yes, please mention in the results that all variables are distributed normally. If not, the reviewer would recommend to do a visual inspection of the variable distribution, and present variable as median with interquartile range when widely deviating from normal.

Minor remarks

- To facilitate reading and to better understand the abstract, the reviewer recommends to rewrite the first sentence of the conclusion of the abstract. ‘We observed a lower prevalence of OHCA in our region as compared to previous literature’ (page 4, line 57) insinuates a lower overall OHCA prevalence, while this prevalence is calculated in a population of ACS patients.

- Besides, please clarify that a lower prevalence of OHCA in the Gulf region is found as compared to previous literature from other regions (e.g., USA [3, 22], Denmark[21]). One might misinterpret that the prevalence of OHCA in the Gulf region has decreased over the years since the previous estimation of OHCA prevalence in the same region.

- The authors state that one-year follow-up data was available for 3 out of the 7 registries (page 6, Line 91). However, according to S1Table the number of registries with a 1-year follow-up comes to a total of 4 out of the 7 registries. Please clarify this.

- Please add the absolute number of OHCA cases in all registries in Table 4 (page 23, second column).

- In the reviewer’s opinion, it is not surprising that STEMI is found to be a predictor for OHCA in an ACS registry. The authors state in the discussion (page 21, line 212) that this finding is helpful in the risk-stratification in young adults with ACS for OHCA. Please explain the clinical relevance of this finding.

- Please remove the redundant number “9” in Table 4, page 27, seventh column (column Predictors of OHCA, row Fordyce et al.,[2] 2016).

- The percentage of primary PCI is extremely low (20-30%), given the fact that the majority of OHCA’s was based on STEMI. Please comment.

Reviewer #2: The current manuscript summarizes findings of an observational study pooling data from 7 prospective registries for acute coronary syndrome in the Gulf region. The primary goal was to investigate outcomes of patients suffering out-of-hospital cardiac arrest. Since a large proportion of patients admitted with OHCA is below the age of 50 years, the authors to specifically target this patient group and compare against older patients admitted after OHCA. The recruitment period of the individual registries was from 2005 to 2017 and only 3 of 7 had follow up data up to 1 year available.

Overall, 31,620 ACS patients were included in this analysis, of whom 34.3% were considered young (below 50 years of age). 611 (1.93%) of cases presented after OHCA in the entire cohort. Overall, 56.6% of OHCA patients survived and were discharged from the hospitals. The prevalence of OHCA was similar between young and old patients. Young adults were predominantly males presenting with first-time STEMI (96.8%). Younger patients with OHCA had lower prevalence of comorbidities. Younger adults received more often primary PCI and had lower mortality compared to older adults with OHCA.

The manuscript is well written and the topic remains timely and interesting as there is chronic lack of data in this field. The authors report on a specific subgroup of patients, which is noteworthy and strength of the manuscript. Nevertheless, the study remains highly descriptive and the findings (younger OHCA patients presenting with STEMI as the first-time manifestation of CAD) are not necessarily novel. As such, the authors should try to provide greater level of granularity with respect to out-of-hospital, procedural and post-procedural determinants of poor outcome. It is understood that only 3 out of 7 registries provided follow-up data; nevertheless, the authors could make an attempt to derive predictors of mortality in these prespecified subgroups. Impact of therapeutic strategies may be of interest in this regard. Moreover, nothing is reported on overall outcome of these patients, i.e. neurological outcome, heart failure etc. Given the fact that almost 50% of OHCA patients survived, sufficient data should be available to investigate these important questions.

Major points:

Methods, page 6, variables, the reviewer is missing important known determinants of outcome among OHCA arrest patients such as witnessed arrest, arrest during daytime or nighttime, bystander resuscitation etc. What was the first documented rhythm in these patients? What was the first pH, lactate etc in these patients. Greater level of granularity is required.

Results, page 7, the overall survival rate of 56.6% is extremely high for OHCA patients. This point needs to be discussed with greater focus on potential selection bias (i.e. very young patients, mostly STEMI patients)

Supplemental file, the individual registries only included a relatively low proportion of ACS patients, i.e. between 3.9 to 25.8%. Against the notion of the authors that patients were recruited from ACS registries, what was the reason for enrolment in these registries? This also explains the low number of OHCA patients (1.93%), which resembles a dual selection bias, one arising from the registry itself and the second is the survival bias (patients dying of SCD will never present to the emergency department). This point needs to be addressed.

Predictors of OHCA, the authors should try to discuss the smokers’ paradox in a more balanced way. Significant literature is available on this topic and very likely, this paradox does not even exist and is a perfect example of selection bias in these kind of analysis.

The authors should consider time-dependent analysis given the long recruitment phase from 2005-2017

The authors fail to discuss most contemporary literature on OHCA patients presenting with ACS. A number of major landmark studies were published (i.e. Tomahawk and others) that are not even mentioned.

6. PLOS authors have the option to publish the peer review history of their article (what does this mean?). If published, this will include your full peer review and any attached files.

Reviewer #1: **Yes: **Niels van Royen

Reviewer #2: No

---

## [Author Response · Author response to Decision Letter 0]

30 Mar 2023

March 30, 2023

RE: “Characteristics and predictors of out-of-hospital cardiac arrest in young adults hospitalized with acute coronary syndrome: A retrospective cohort study of 30,000 patients in the Gulf region.” (PONE-D-22-33714)

Dear Dr. AlSaif,

The authors would like to thank the editor and reviewers for taking the time to appraise our manuscript. We have revised the manuscript based on the reviewers’ suggestions. The comments are in bold, and our responses follow. 

Reviewer #1: 

The authors aim to describe the characteristics and predictors of OHCA in young adults (aged below 50 years) hospitalized with ACS. This is a retrospective cohort study using data from 7 prospective ACS registries in the Gulf region. The main findings are:

- The prevalence of OHCA in an ACS population in the Gulf region is lower as compared to previous literature.

- Young OHCA patients with ACS are more likely to be male and have STEMI with low prevalence of traditional cardiovascular risk factors, compared to older adults with OHCA.

- OHCA is the sentinel event of CAD (i.e., the presenting symptom) in the majority of young adults (29/41 (70.73%)).

Their study is unique because it examines young adults separately. However, there are some remarks to be made on this manuscript.

1- It is stated that “causes of OHCA/SCD are distinctly different in those younger than 50 where CAD is not the most common cause.[18]” This is true for patients below 40 years of age, but above 40 also CAD and associated acute coronary syndrome is the main cause of OHCA (Empana et al, JACC, 2022). Please provide data on patients <40 years.

Thank you for valuable suggestion. A new analysis for patients <40 (referred to as “very young”) was added to the manuscript.

• Added S2 Table as a new supplemental information file.

• Page 7, Line 125: Added “Additional analysis for “very young” patients was performed, where the age cutoff was <40 years instead of <50 years given that CAD was found to be the main cause of OHCA in some studies.[14]”

• Page 21, Line 192: Added “Sensitivity analysis: S2 Table shows the characteristics and predictors of “very young” OHCA patients (<40 years). “Very young” OHCA patients had similar features to what we described in young patients (<50 years), with the majority being male [53/54 (98.15%)] and presenting with STEMI [46/54 (85.19%)]. However, “very young” patients had an even lower prevalence of CAD risk factors such as hypertension [4/54 (7.41%)] and diabetes mellitus [6/54 (11.11%)].”

2- The third main finding as listed above (OHCA is the sentinel event of CAD in the majority of young adults) is based on a sample size of 41 patients, see page 10 Table 1. Since this is described as one of the main results of the paper, please elaborate further on this small sample size. It is striking that for the other variables in Table 1 page 10 (subset ‘presentation data’) there is an almost complete sample size of 188 OHCA patients.

- Please include this in the future recommendations as well.

Thank you for your thoughtful comment. Indeed, the denominator for this variable is lower than other variables due to missing data. However, we note that this was statistically significant (p=0.0042) with a large effect size (-0.2160) suggesting a true important difference. We agree however that a larger sample size would have provided a more precise estimate and we included this in the future recommendation as suggested. 

• Page 24, Line 272: added “Also, further studies with a focus on the presentation of OHCA as the sentinel event of CAD in young patients using a larger sample size need to be performed in order to confirm our findings.”

3- In the statistical analysis section (page 6 line 106) it is described that all variables are reported as means with standard deviations. Are the variables tested for normal distribution? If yes, please mention in the results that all variables are distributed normally. If not, the reviewer would recommend to do a visual inspection of the variable distribution, and present variable as median with interquartile range when widely deviating from normal.

We have performed visual analysis with histograms and Q-Q plots, and determined that the following variables are not normally distributed, and therefore, median with interquartile range was reported instead of mean with SD:

• Table 1 and Table 2: Symptoms to hospital arrival time updated to median with interquartile range.

Minor remarks

4- To facilitate reading and to better understand the abstract, the reviewer recommends to rewrite the first sentence of the conclusion of the abstract. ‘We observed a lower prevalence of OHCA in our region as compared to previous literature’ (page 4, line 57) insinuates a lower overall OHCA prevalence, while this prevalence is calculated in a population of ACS patients.

We updated the sentence as suggested.

• Page 4, Line 56: modified the sentence to “We observed a lower prevalence of OHCA in ACS patients in our region as compared to previous literature from other regions.”

5- Besides, please clarify that a lower prevalence of OHCA in the Gulf region is found as compared to previous literature from other regions (e.g., USA [3, 22], Denmark[21]). One might misinterpret that the prevalence of OHCA in the Gulf region has decreased over the years since the previous estimation of OHCA prevalence in the same region.

We updated the sentence as per the previous comment.

• Page 4, Line 56: modified the sentence to “We observed a lower prevalence of OHCA in ACS patients in our region as compared to previous literature from other regions.”

6- The authors state that one-year follow-up data was available for 3 out of the 7 registries (page 6, Line 91). However, according to S1Table the number of registries with a 1-year follow-up comes to a total of 4 out of the 7 registries. Please clarify this.

Thank you for highlighting this mistake. We have corrected the wrong statement.

• Page 6, Line 89: corrected to “One-year follow-up data was available for 4 out of the 7 registries.”

7- Please add the absolute number of OHCA cases in all registries in Table 4 (page 23, second column).

This was added to Table 4 as per the reviewer’s recommendation.

• Table 4: Added in third column.

8- In the reviewer’s opinion, it is not surprising that STEMI is found to be a predictor for OHCA in an ACS registry. The authors state in the discussion (page 21, line 212) that this finding is helpful in the risk-stratification in young adults with ACS for OHCA. Please explain the clinical relevance of this finding.

We agree with the reviewer that STEMI being a predictor is not a novel/surprising finding. Nonetheless, we believe that, given the different characteristics of our population, confirming this finding in our population is important.

Risk stratification of ACS for OHCA can be helpful for research and clinical purposes. It can help select “high-risk” patients to test any intervention to improve OHCA outcomes. Clinically, it can help better utilize resources especially in large countries with limited access to EMS. For example, predictors of OHCA can be used in deciding who gets priority for early transfer to tertiary care centers/OHCA centers of excellence when presenting with ACS. We have updated our manuscript to elaborate on this:

• Page 23, line 258: added “This is helpful in risk-stratifying young adults with ACS for OHCA where these criteria can be used to select higher risk patients to examine future interventions and to prioritize them for more urgent care.” 

9- Please remove the redundant number “9” in Table 4, page 27, seventh column (column Predictors of OHCA, row Fordyce et al.,[2] 2016).

This was corrected.

• Table 4: Modified, seventh column, row Fordyce et al.

10- The percentage of primary PCI is extremely low (20-30%), given the fact that the majority of OHCA’s was based on STEMI. Please comment.

This is an important observation. Indeed, low rates of revascularization has been consistently shown in previous studies from the Gulf region.[1-7] For example, the “Gulf RACE 1” registry reported that only 7% of ACS patients in their registry received PCI.[7] The rate improved in “Gulf RACE 2” to 21%, however, it’s still quite lower than expected.[2] The overall low percentage is thought to be due to the lack of catheterization laboratories in most of the hospitals in the region during the early period of the registries.[1-3]This is supported by the time-dependent analysis we performed in the current study where the rate of primary PCI in STEMI was 8.15% (S3A Table) in patients enrolled before 2011 and 37.33% in patients enrolled after that (S4A Table). This improvement is, in most part, a reflection of the increase in the number of PCI-capable hospitals in the region. 

• Page 21, Line 199: Added “S3 and S4 Tables report time-dependent analysis before and after 2011, respectively. The total prevalence of OHCA before 2011 was 262/7,310 (3.58%) which was higher after 2011[349/3,537 (9.87%)]. However, the proportion of young adults with OHCA remained similar for both time periods [79/262 (30.15%)] before 2011 and [109/349 (31.23%)] after 2011. Differences were also seen regarding reperfusion therapy. For example, the rate of primary PCI in STEMI patients improved over time, from 8.15% before 2011, to 37.33% after 2011.”

Reviewer #2: 

1-The manuscript is well written and the topic remains timely and interesting as there is chronic lack of data in this field. The authors report on a specific subgroup of patients, which is noteworthy and strength of the manuscript. Nevertheless, the study remains highly descriptive and the findings (younger OHCA patients presenting with STEMI as the first-time manifestation of CAD) are not necessarily novel. As such, the authors should try to provide greater level of granularity with respect to out-of-hospital, procedural and post-procedural determinants of poor outcome. It is understood that only 3 out of 7 registries provided follow-up data; nevertheless, the authors could make an attempt to derive predictors of mortality in these prespecified subgroups. Impact of therapeutic strategies may be of interest in this regard. Moreover, nothing is reported on overall outcome of these patients, i.e. neurological outcome, heart failure etc. Given the fact that almost 50% of OHCA patients survived, sufficient data should be available to investigate these important questions.

Thank you for your thoughtful review. We have derived predictors of in-hospital and 1-year mortality. In addition, we have described more in-hospital outcomes and complications. Unfortunately, the only long-term outcome collected is mortality.

• Added supplemental S5 Table.

• Page 7, line 132: Added “Moreover, we performed univariate and multivariate logistic regression analysis for potential predictors of in-hospital and 1-year mortality among the entire ACS sample.”.

• Page 21, line 206: Added “Predictors of mortality: S5A-B Tables show the predictors of in-hospital and 1-year mortality in ACS patients. OHCA was found to be an independent predictor of in-hospital but not 1-year mortality [OR=2.673 (95% CI 1.271–5.620) for young adults, OR=3.194 (95% CI 1.872–5.450) for older adults; p <.0001] and [OR=1.547 (95% CI 0.246– 9.744) for young adults, OR=0.816 (95% CI 0.114– 5.823) for older adults; p = 0.6394], respectively. There was no interaction between OHCA and young age for both in-hospital and 1-year mortality [p = 0.6960] and [p = 0.6394], respectively. Revascularization was protective for in-hospital and 1-year mortality [OR=0.329 (95% CI 0.241–0.448); p <.0001] and [OR=0.674 (95% CI 0.478– 0.948); p = 0.0236], respectively. “

• Table 1: Added “in-hospital complications” variables. 

• Table 2: Added “in-hospital complications” variables. 

Major points:

2-Methods, page 6, variables, the reviewer is missing important known determinants of outcome among OHCA arrest patients such as witnessed arrest, arrest during daytime or nighttime, bystander resuscitation etc. What was the first documented rhythm in these patients? What was the first pH, lactate etc in these patients. Greater level of granularity is required.

We completely agree with the reviewer that these are important variables for a better description of OHCA in general. Unfortunately, these are not available. The focus of these registries was to describe the general characteristics of ACS patients in the region. 

3-Results, page 7, the overall survival rate of 56.6% is extremely high for OHCA patients. This point needs to be discussed with greater focus on potential selection bias (i.e. very young patients, mostly STEMI patients)

Thank you for the opportunity to highlight this important point. The apparent high survival rate is indeed limited by survival bias. Only patients who survived to hospital admission were included. This is consistent with previous similar studies that reported a survival rate of 56% - 71%.[8-11] We have added the survival rate of each study in Table 4, in order to facilitate comparison for readers. In addition, we discussed survival bias in our limitation.

• Table 4: Second column, added survival rate of each study.

• Page 30, line 287: “Third, the apparent high survival rate should be interpreted with cation given the survival bias. Indeed, only patients with ACS who survived to hospital admission were included in these registries which results in a biased estimate of survival.”

4-Supplemental file, the individual registries only included a relatively low proportion of ACS patients, i.e. between 3.9 to 25.8%. Against the notion of the authors that patients were recruited from ACS registries, what was the reason for enrolment in these registries? This also explains the low number of OHCA patients (1.93%), which resembles a dual selection bias, one arising from the registry itself and the second is the survival bias (patients dying of SCD will never present to the emergency department). This point needs to be addressed.

Thank you for the opportunity to clarify this important point. All patients included had ACS. The percentages in the table represent the contribution of each registry to the overall data set. We have modified the table to clarify this point.

• S1 Table: Added the total sample size as the denominator for each registry in the second row. 

5-Predictors of OHCA, the authors should try to discuss the smokers’ paradox in a more balanced way. Significant literature is available on this topic and very likely, this paradox does not even exist and is a perfect example of selection bias in these kind of analysis.

We have updated our manuscript to provide a more detailed overview of the paradox and potential explanations for this finding.

• Page 22, Line 236: Added “It is unclear if “smoking paradox” represents a true phenomenon or a spurious finding. Aune et al reported a systematic review that reviewed the evidence behind “smoking paradox”.[25] They examined 17 studies in total and found that this paradox is not always found, especially in studies using contemporary therapies. Nonetheless, a more recent study utilizing a prospective cohort study of more than 37,000 patients with myocardial infarction found that smokers are younger and have fewer comorbidities.[26] However, even after adjustment for confounders, smokers were found to have a lower in-hospital mortality. It is certainly possible that residual confounding explains this apparent association in this study and others, including ours.”

6-The authors should consider time-dependent analysis given the long recruitment phase from 2005-2017

We have performed a time-dependent analysis as suggested. Patients were divided based on recruitment year into 2 groups: before 2011 and after 2011. A complete analysis was repeated.

• Added: S3 Table (A, B) for patients before 2011.

• Added: S4 Table (A, B) for patients after 2011.

• Page 7, Line 125: Added “We performed sensitivity analysis including time-dependent analysis, with a cutoff year of 2011. This was done as the study had a long recruitment period extending from 2005 to 2017.”

• Page 21, Line 199: Added “S3 and S4 Tables report time-dependent analysis before and after 2011, respectively. The total prevalence of OHCA before 2011 was 262/7,310 (3.58%) which was higher after 2011[349/3,537 (9.87%)]. However, the proportion of young adults with OHCA remained similar for both time periods [79/262 (30.15%)] before 2011 and [109/349 (31.23%)] after 2011. Differences were also seen regarding reperfusion therapy. For example, the rate of primary PCI in STEMI patients improved over time, from 8.15% before 2011, to 37.33% after 2011.”

• Page 30, Line 282: Added “We also performed time-dependent analysis (shown in S3 and S4 Tables), in order to provide data across different time periods.”

7-The authors fail to discuss most contemporary literature on OHCA patients presenting with ACS. A number of major landmark studies were published (i.e. Tomahawk and others) that are not even mentioned.

We have focused on real-life, registry-based data in our discussion and comparisons. As you know, clinical trials have specific inclusion and exclusion criteria that limit their ability to represent characteristics of populations studied. Since the main goal of our study was to describe the characteristics and predictors of OHCA, we elected to avoid including clinical trials as the characteristics of patients included might not represent real-world findings. For example, the study by Tomahawk and others was a trial of OHCA patients to test the potential benefit of revascularization in resuscitated patients without electrocardiographic evidence of ST-segment elevation. Thus, by design, they excluded STEMI patients precluding any comparison with our study when it comes to “type of ACS”. In addition, they excluded patients younger than 30 years of age which affected the average age of their population.

Editors:

Journal Requirements:

1- Please ensure that your manuscript meets PLOS ONE's style requirements, including those for file naming. The PLOS ONE style templates can be found at 

We have ensured that the manuscript follows the PLOS ONE's style requirements.

2- In the Methods section of your revised manuscript, please include the names of all of the IRBs that approved the seven registries

We have updated the ethics statement.

• Page 6, Line 101: The statement was modified to “The study was exempted from the submission for review by the Institutional Review Board (IRB), College of Medicine, King Saud University, Riyadh, Saudi Arabia. As the data used in this study consisted of secondary published data that has been completely de-identified with no direct or indirect identifiers available in the database and from the registries that were individually approved through the authors by their respective IRB institutions as follows: [7-13]

RACE 1, RACE 2, RACE 3, STARS, and SPACE: The registries were approved by the IRB of College of Medicine, King Saud University, Riyadh, Saudi Arabia, with additional approval by each participating hospital.

COAST: The registry was approved by the institutional review board of each participating hospital (29 hospitals) in each of the following countries: Kuwait, Bahrain, Oman, and the United Arabic Emirates.

REPERFUSE: The registry was approved by the Kuwait Ministry of Health’s ethics committee for the protection of human subjects.”

3- Thank you for including your ethics statement: "The data used in this study were obtained from 7 registries which were all approved by their respective institution review boards". 

a. For studies reporting research involving human participants, PLOS ONE requires authors to confirm that this specific study was reviewed and approved by an institutional review board (ethics committee) before the study began. Please provide the specific name of the ethics committee/IRB that approved your study, or explain why you did not seek approval in this case.

b. Please provide additional details regarding participant consent. In the ethics statement in the Methods and online submission information, please ensure that you have specified (1) whether consent was informed and (2) what type you obtained (for instance, written or verbal, and if verbal, how it was documented and witnessed). If your study included minors, state whether you obtained consent from parents or guardians. If the need for consent was waived by the ethics committee, please include this information.

This was modified as a part of the updated ethics statement mentioned in the comment above.

4- In your Data Availability statement, you have not specified where the minimal data set underlying the results described in your manuscript can be found. PLOS defines a study's minimal data set as the underlying data used to reach the conclusions drawn in the manuscript and any additional data required to replicate the reported study findings in their entirety. All PLOS journals require that the minimal data set be made fully available. For more information about our data policy, please see http://journals.plos.org/plosone/s/data-availability.

The study’s dataset has been uploaded to Figshare.com, with the following DOI: 

https://dx.doi.org/10.6084/m9.figshare.22340509

References:

1. Zubaid M, Rashed WA, Al-Khaja N, Almahmeed W, Al-Lawati J, Sulaiman K, et al. Clinical presentation and outcomes of acute coronary syndromes in the gulf registry of acute coronary events (Gulf RACE). Saudi Med J. 2008;29(2):251-5. PubMed PMID: 18246236.

2. Alhabib KF, Sulaiman K, Al-Motarreb A, Almahmeed W, Asaad N, Amin H, et al. Baseline characteristics, management practices, and long-term outcomes of Middle Eastern patients in the Second Gulf Registry of Acute Coronary Events (Gulf RACE-2). Annals of Saudi Medicine. 2012;32(1):9-18. doi: 10.5144/0256-4947.2012.9.

3. Alhabib KF, Sulaiman K, Al Suwaidi J, Almahmeed W, Alsheikh-Ali AA, Amin H, et al. Patient and System-Related Delays of Emergency Medical Services Use in Acute ST-Elevation Myocardial Infarction: Results from the Third Gulf Registry of Acute Coronary Events (Gulf RACE-3Ps). PLOS ONE. 2016;11(1):e0147385. doi: 10.1371/journal.pone.0147385.

4. Zubaid M, Rashed W, Alsheikh-Ali AA, Garadah T, Alrawahi N, Ridha M, et al. Disparity in ST-segment Elevation Myocardial Infarction Practices and Outcomes in Arabian Gulf Countries (Gulf COAST Registry). Heart Views. 2017;18(2):41-6. doi: 10.4103/HEARTVIEWS.HEARTVIEWS_113_16. PubMed PMID: 28706594; PubMed Central PMCID: PMCPMC5501027.

5. Alhabib KF, Kinsara AJ, Alghamdi S, Al-Murayeh M, Hussein GA, Alsaif S, et al. The first survey of the Saudi Acute Myocardial Infarction Registry Program: Main results and long-term outcomes (STARS-1 Program). PLOS ONE. 2019;14(5):e0216551. doi: 10.1371/journal.pone.0216551.

6. Alhabib KF, Hersi A, Alfaleh H, Alnemer K, Alsaif S, Taraben A, et al. Baseline characteristics, management practices, and in-hospital outcomes of patients with acute coronary syndromes: Results of the Saudi project for assessment of coronary events (SPACE) registry. J Saudi Heart Assoc. 2011;23(4):233-9. Epub 20110601. doi: 10.1016/j.jsha.2011.05.004. PubMed PMID: 23960654; PubMed Central PMCID: PMCPMC3727434.

7. Zubaid M, Khraishah H, Alahmad B, Rashed W, Ridha M, Alenezi F, et al. Efficacy and Safety of Pharmacoinvasive Strategy Compared to Primary Percutaneous Coronary Intervention in the Management of ST-Segment Elevation Myocardial Infarction: A Prospective Country-Wide Registry. Annals of Global Health. 2020;86(1):13. doi: 10.5334/aogh.2632.

8. Fordyce CB, Wang TY, Chen AY, Thomas L, Granger CB, Scirica BM, et al. Long-Term Post-Discharge Risks in Older Survivors of Myocardial Infarction With and Without Out-of-Hospital Cardiac Arrest. J Am Coll Cardiol. 2016;67(17):1981-90. doi: 10.1016/j.jacc.2016.02.044. PubMed PMID: 27126525.

9. Kontos MC, Scirica BM, Chen AY, Thomas L, Anderson ML, Diercks DB, et al. Cardiac arrest and clinical characteristics, treatments and outcomes among patients hospitalized with ST-elevation myocardial infarction in contemporary practice: A report from the National Cardiovascular Data Registry. Am Heart J. 2015;169(4):515-22.e1. Epub 20150203. doi: 10.1016/j.ahj.2015.01.010. PubMed PMID: 25819858.

10. Dawson LP, Dinh D, Duffy S, Brennan A, Clark D, Reid CM, et al. Short- and long-term outcomes of out-of-hospital cardiac arrest following ST-elevation myocardial infarction managed with percutaneous coronary intervention. Resuscitation. 2020;150:121-9. Epub 20200321. doi: 10.1016/j.resuscitation.2020.03.003. PubMed PMID: 32209377.

11. Kosugi S, Shinouchi K, Ueda Y, Abe H, Sogabe T, Ishida K, et al. Clinical and Angiographic Features of Patients With Out-of-Hospital Cardiac Arrest and Acute Myocardial Infarction. J Am Coll Cardiol. 2020;76(17):1934-43. doi: 10.1016/j.jacc.2020.08.057. PubMed PMID: 33092729.

---

## [Decision Letter · Decision Letter 1]

26 Apr 2023

PONE-D-22-33714R1Characteristics and predictors of out-of-hospital cardiac arrest in young adults hospitalized with acute coronary syndrome: A retrospective cohort study of 30,000 patients in the Gulf region.PLOS ONE

Dear Dr. Alqarawi,

Thank you for submitting your manuscript to PLOS ONE. After careful consideration, we feel that it has merit but does not fully meet PLOS ONE’s publication criteria as it currently stands. Therefore, we invite you to submit a revised version of the manuscript that addresses the points raised during the review process.

We look forward to receiving your revised manuscript.

Kind regards,

Shukri AlSaif

Academic Editor

PLOS ONE

Journal Requirements:

Additional Editor Comments (if provided):

Reviewers' comments:

Reviewer's Responses to Questions

**Comments to the Author**

1. If the authors have adequately addressed your comments raised in a previous round of review and you feel that this manuscript is now acceptable for publication, you may indicate that here to bypass the “Comments to the Author” section, enter your conflict of interest statement in the “Confidential to Editor” section, and submit your "Accept" recommendation.

Reviewer #1: (No Response)

Reviewer #2: (No Response)

2. Is the manuscript technically sound, and do the data support the conclusions?

Reviewer #1: Yes

Reviewer #2: Yes

3. Has the statistical analysis been performed appropriately and rigorously? 

Reviewer #1: Yes

Reviewer #2: Yes

4. Have the authors made all data underlying the findings in their manuscript fully available?

Reviewer #1: Yes

Reviewer #2: Yes

5. Is the manuscript presented in an intelligible fashion and written in standard English?

Reviewer #1: Yes

Reviewer #2: Yes

6. Review Comments to the Author

Reviewer #1: The authors are thanked for their consideration, revisions and additional analyses. To the reviewer’s opinion, the adjustments that are made benefit the manuscript. The manuscript is suitable for publication, but the reviewer recommends to consider the request addressed below.

Although the conclusion that OHCA is a sentinel event of CAD appears reasonable from a clinical perspective, the reviewer remains hesitant about the data on which this conclusion is drawn. Is it possible that the missing data introduce a bias for this variable? The authors are asked to provide an explanation for the missing data. Specifically, the authors are asked why this variable is only available for 41 out of the 188 OHCA patients, and how this may impact the obtained results. Furthermore, the authors are requested to include this in the limitations section and to attenuate the concluding statement of this finding.

Reviewer #2: the authors have appropriately addressed most of my concerns. I would only like to see a direct comparison of the time-dependent analysis based on the two time-periods with regards to mortality in one single table. This reviewer believes that outcomes are substantially dependent on treatment modalities and these differ quite substantially among time-periods chosen (20.7% vs. 16.5%).

7. PLOS authors have the option to publish the peer review history of their article (what does this mean?). If published, this will include your full peer review and any attached files.

Reviewer #1: **Yes: **Niels van Royen

Reviewer #2: No

---

## [Author Response · Author response to Decision Letter 1]

8 May 2023

May 8, 2023

RE: “Characteristics and predictors of out-of-hospital cardiac arrest in young adults hospitalized with acute coronary syndrome: A retrospective cohort study of 30,000 patients in the Gulf region.” (PONE-D-22-33714R1)

Dear Dr. AlSaif,

The authors would like to thank the editor and reviewers again for reviewing the revised manuscript. We have updated the manuscript based on the reviewers’ suggestions. The comments are in bold, and our responses follow. 

Reviewer #1: 

The authors are thanked for their consideration, revisions and additional analyses. To the reviewer’s opinion, the adjustments that are made benefit the manuscript. The manuscript is suitable for publication, but the reviewer recommends to consider the request addressed below.

Although the conclusion that OHCA is a sentinel event of CAD appears reasonable from a clinical perspective, the reviewer remains hesitant about the data on which this conclusion is drawn. Is it possible that the missing data introduce a bias for this variable? The authors are asked to provide an explanation for the missing data. Specifically, the authors are asked why this variable is only available for 41 out of the 188 OHCA patients, and how this may impact the obtained results. Furthermore, the authors are requested to include this in the limitations section and to attenuate the concluding statement of this finding.

The reason for the “missing data” is that this variable was only available in some of the registries that contributed to the data set. It is certainly possible that this may have introduced a selection bias. We have added this explanation, discuss it in the limitation section, and revised our conclusion:

• Page 23, Line 268: added “It should be noted that this finding is limited by the smaller sample size available for this variable, given that it was not collected in all registries.”

• Page 29, Line 287: added “Fourth, our finding of OHCA as the sentinel event of CAD was based on a smaller sample compared to the other variables, which affects the precision of this estimate.”

• Page 30, Line 298: modified the conclusion sentence to “Moreover, OHCA was the sentinel event of CAD in many young adults.”

Reviewer #2: 

the authors have appropriately addressed most of my concerns. I would only like to see a direct comparison of the time-dependent analysis based on the two time-periods with regards to mortality in one single table. This reviewer believes that outcomes are substantially dependent on treatment modalities and these differ quite substantially among time-periods chosen (20.7% vs. 16.5%).

We thank the reviewer for their valuable feedback. A new supplemental table for the results of time dependent mortality rates between the two time periods was added as suggested (Table S5C). Also, we added a summary of the results of this table in the results section of the manuscript.

• Added S5C Table as a new supplemental information file.

• Page 21, Line 210: added “S5C Table compares the mortality rates of OHCA patients before and after 2011. We observed higher mortality before 2011, albeit not statistically significant.”

Journal Requirements:

We have reviewed the references and ensured that it is complete and correct.

---

## [Editor Report · Decision Letter 2]

9 May 2023

Characteristics and predictors of out-of-hospital cardiac arrest in young adults hospitalized with acute coronary syndrome: A retrospective cohort study of 30,000 patients in the Gulf region.

PONE-D-22-33714R2

Dear Dr. Alqarawi,

We’re pleased to inform you that your manuscript has been judged scientifically suitable for publication and will be formally accepted for publication once it meets all outstanding technical requirements.

Kind regards,

Shukri AlSaif

Academic Editor

PLOS ONE

---

## [Editor Report · Acceptance letter]

17 May 2023

PONE-D-22-33714R2 

Characteristics and predictors of out-of-hospital cardiac arrest in young adults hospitalized with acute coronary syndrome: A retrospective cohort study of 30,000 patients in the Gulf region.  

Dear Dr. Alqarawi:

I'm pleased to inform you that your manuscript has been deemed suitable for publication in PLOS ONE. Congratulations! Your manuscript is now with our production department. 

Kind regards, 

on behalf of

Dr. Shukri AlSaif 

Academic Editor

PLOS ONE